The EMBO Journal (2013) 32, 1155–1167
www.embojournal.org

# Premature Cdk1/Cdc5/Mus81 pathway activation induces aberrant replication and deleterious crossover

## Barnabas Szakal and Dana Branzei*

Department of Molecular Oncology, Fondazione IFOM, Istituto FIRC di Oncologia Molecolare, Milan, Italy

**The error-free DNA damage tolerance (DDT) pathway is crucial for replication completion and genome integrity. Mechanistically, this process is driven by a switch of templates accompanied by sister chromatid junction (SCJ) formation. Here, we asked if DDT intermediate processing is temporarily regulated, and what impact such regulation may have on genome stability. We find that persistent DDT recombination intermediates are largely resolved before anaphase through a G2/M damage checkpoint-independent, but Cdk1/Cdc5-dependent pathway that proceeds via a previously described Mus81-Mms4-activating phosphorylation. The Sgs1-Top3- and Mus81-Mms4-dependent resolution pathways occupy different temporal windows in relation to replication, with the Mus81-Mms4 pathway being restricted to late G2/M. Premature activation of the Cdk1/Cdc5/Mus81 pathway, achieved here with phosphomimetic Mms4 variants as well as in S-phase checkpoint-deficient genetic backgrounds, induces crossover-associated chromosome translocations and precocious processing of damage-by-pass SCJ intermediates. Taken together, our results underscore the importance of uncoupling error-free versus erroneous recombination intermediate processing pathways during replication, and establish a new paradigm for the role of the DNA damage response in regulating genome integrity by controlling crossover timing.**

*The EMBO Journal* (2013) **32,** 1155–1167. doi:10.1038/emboj.2013.67; Published online 26 March 2013
*Subject Categories:* cell cycle; genome stability & dynamics
*Keywords*: ATR/Mec1 replication checkpoint; Cdk1/Cdc5 kinases; chromosome translocations; mitotic cell cycle; template switching

## Introduction

Maintenance of genome integrity requires an accurate interplay between replication, repair and cell-cycle transitions. Replication fork blocks, as well as DNA lesions contained in gaps behind replication forks, are bypassed via DNA damage tolerance (DDT) mechanisms that either involve specialized translesion synthesis polymerases or homologous recombi-

nation (HR) mechanisms (Lopes *et al*, 2006; Mizuno *et al*, 2009; Paek *et al*, 2009; Daigaku *et al*, 2010; Karras and Jentsch, 2010). The latter strategy, known as error-free DDT, involves the transient formation in the proximity of replication forks of sister chromatid junction (SCJ) intermediates, whose processing/resolution during replication is largely driven by Sgs1-Top3 and Ubc9-, Mms21-Smc5-6-mediated SUMOylation (Liberi *et al*, 2005; Branzei *et al*, 2006, 2008; Sollier *et al*, 2009). Genetic characterization of template switching established that it occurs via a specialized recombination mechanism involving the highly conserved eukaryotic pathway of postreplicative repair (PRR), dependent on *RAD6-RAD18-RAD5* and PCNA polyubiquitylation, as well as a subset of HR factors (Branzei *et al*, 2008; Minca and Kowalski, 2010; Vanoli *et al*, 2010; Karras *et al*, 2012). The Sgs1 helicase, homologue of human BLM, which is mutated in cancer-prone Bloom syndrome patients, functions together with Top3, downstream of PCNA polyubiquitylation, in the error-free DDT pathway by processing replication-associated damage-bypass SCJs (Branzei *et al*, 2008; Karras and Jentsch, 2010).

Proteins involved in the error-free branch of PRR or DDT are thought to function as tumour suppressors, but the links of this pathway to tumorigenesis have remained elusive. Sgs1/BLM, together with Top3, can resolve/dissolve recombination intermediates such as double Holliday Junctions (dHJs), as well as template switch SCJ intermediates, to non-crossover products (Ira *et al*, 2003; Wu and Hickson, 2003; Liberi *et al*, 2005; Robert *et al*, 2006; Branzei *et al*, 2008), that is, with no exchange between the donor and recipient DNA sequences. Mus81-Mms4/Eme1, Slx1-Slx4, and Yen1/Gen1 structure-specific endonucleases also process *in vitro* single or double HJs, and during double-strand break (DSB) repair their action leads to crossover outcome (exchange of DNA sequences between the donor and the recipient) (Ip *et al*, 2008; Ho *et al*, 2010; Wechsler *et al*, 2011; Munoz-Galvan *et al*, 2012). Recently, two of these nucleases, Mus81-Mms4 and Yen1 were shown to be activated upon mitotic entry (Matos *et al*, 2011), while other genetic data implicated both Sgs1-Top3 and Mus81-Mms4 in replication restart or in processing replication structures (Fabre *et al*, 2002; Hanada *et al*, 2007; Osman and Whitby, 2007; Kang *et al*, 2010). Clearly, the individual contribution of various nucleases and helicases, as well as the mechanisms controlling how replication-born DDT recombination intermediates are processed and when, remains to be elucidated.

The damage checkpoint pathway and cell-cycle kinases, such as cyclin-dependent kinases (CDKs) and polo-like kinases, play key roles in coordinating DNA repair with cell-cycle transitions (Sanchez *et al*, 1999; Bartek and Lukas, 2007; Harper and Elledge, 2007; Branzei and Foiani, 2008). Central components of the checkpoint machinery are the apical phosphoinositide 3-kinase-related kinases, Mec1

*Corresponding author. Department of Molecular Oncology, Fondazione IFOM, Istituto FIRC di Oncologia Molecolare, IFOM-IEO Campus, Via Adamello 16, Milano 20139, Italy. Tel.: +39 02574303259; Fax: +39 02574303231; E-mail: dana.branzei@ifom.eu

and Tel1 in budding yeast (ATR and ATM in human cells, respectively). Rad53 phosphorylation by Mec1-Ddc2 (ATR-ATRIP) is crucial for triggering the replication checkpoint response. Furthermore, when cells incur DNA damage, they activate checkpoint mechanisms that result in S-phase slow down and prolonged G2/M arrest, thereby allowing time for DNA repair (Paulovich and Hartwell, 1995; Tercero and Diffley, 2001; Sancar et al, 2004). Recent work suggested a role for Mec1 and Rad53 proteins in promoting error-free DDT by template switching, but the mechanism involved remains elusive (Liberi et al, 2005; Gangavarapu et al, 2011).

In this work, we investigated if DDT recombination intermediate processing/resolution undergoes cell-cycle regulation. We uncovered that a damage checkpoint-independent, but a Cdk1- and Cdc5-dependent pathway promotes persistent DDT recombination intermediate processing in G2/M. This late resolution proceeds largely before anaphase and involves the Mus81-Mms4 endonuclease and its activation via the previously described Mms4 phosphorylation (Matos et al, 2011; Gallo-Fernandez et al, 2012). We obtained evidence that the Mus81-Mms4 resolution activity is not just potentiated via phosphorylation, but restricted to late G2/M under physiological conditions, and further asked on the consequences of temporarily deregulating the Cdk1/Cdc5/Mus81 pathway activation on genome integrity. We identified here that Mus81-Mms4-dependent resolution is crossover prone in outcome, and its precocious activation leads to faulty replication of damaged templates, as well as to formation of deleterious crossovers associated with chromosome translocations in unperturbed conditions. Although several mechanisms may independently potentiate the temporal restriction of the Mus81 pathway to late G2/M, we identified here that the premature activation of the Cdk1/Cdc5/Mus81 pathway underlies the genome rearrangements and replication defects of S-phase checkpoint mutants, thus explaining the so far elusive role of ATR/Mec1 in promoting error-free DDT, as well as a mechanism by which error-free DDT defects lead to tumorigenesis.

## Results

### DDT recombination intermediate resolution in mitosis is damage checkpoint independent, but Cdk1 and Cdc5 dependent

Error-free DDT intermediates consist primarily of transient SCJs forming in the proximity of replication forks and then dissolved by the action of Sgs1-Top3 (Liberi et al, 2005; Branzei et al, 2008). To address how DDT intermediate resolution proceeds when delayed to G2/M, we followed by 2D gel electrophoresis the fate of replication-derived damage-bypass SCJs accumulating in sgs1 mutant cells or other genetic backgrounds that influence SCJ resolution in S phase. We found that most of the SCJs formed during replication are processed in G2/M before anaphase (Supplementary Figures S1A and B and see below). We could further rule out that the disappearance of the damage-bypass SCJs is due to their branch migration to nearby locations, as at the time when most of these intermediates disappeared from ARS305, they were also not anymore visible in the flanking regions, up to 30 kb apart from ARS305 (Supplementary Figure S1B). Thus, a late recombina-

tion intermediate processing pathway (hereafter referred to as 'late resolution') is activated before anaphase to deal with persistent recombination/DDT intermediates.

We then investigated the regulatory activities facilitating the late resolution. In response to DNA damage, the ATM/ATR checkpoint kinases phosphorylate a vast network of proteins (Matsuoka et al, 2007; Smolka et al, 2007; Chen et al, 2010), and influence cell cycle, DNA repair and DDT responses (Branzei and Foiani, 2008; Ciccia and Elledge, 2010). In budding yeast, Mec1-Ddc2 and Rad53 are essential for cell viability and are involved in both the replication and the G2/M damage checkpoints. To address their role in the late resolution, we established conditional mutants with the help of which we could timely deplete the apical checkpoint kinases following template switch intermediate formation in S phase. This was necessary because constitutive checkpoint inactivation impairs SCJ formation in S phase (Liberi et al, 2005). We established a DDC2 conditional allele (Tc-DDC2-AID) in which Ddc2 translation is prevented upon addition of tetracycline (Kotter et al, 2009), and its degradation is further induced by addition of auxin (Nishimura et al, 2009). This combination of inducible conditional systems leads to markedly reduced viability (as would be expected for Ddc2-depleted cells), and a rapid decrease in Ddc2 levels when cells are treated with tetracycline and auxin (Figure 1A). Following MMS treatment, sgs1 Tc-DDC2-AID cells were allowed to recover in normal media until they reached G2/M and then Ddc2 depletion was induced. Ddc2 depletion occurred efficiently, but did not impair the late resolution (see Figure 1A). To exclude that the lack of effect caused by Ddc2 depletion is due to redundancy between Mec1-Ddc2 and Tel1 pathways, we further combined TEL1 deletion with Ddc2 conditional inactivation. As expected, Rad53 activation was drastically reduced under these conditions, but the late resolution still occurred with normal kinetics (Figure 1A). Thus, the G2/M damage checkpoint pathway controlled by Mec1-Ddc2 and Tel1 is not required for the late resolution.

As mitotic CDK activity has been linked to recombination intermediate processing in S. pombe (Caspari et al, 2002), and the peak of mitotic Cdk1 activity coincides with the time window in which persistent DDT intermediates are resolved (Figure 1A; Supplementary Figure S1B), we examined the requirement for Cdk1 activity in this process. In budding yeast, a single CDK catalytic subunit, Cdk1, encoded by the essential gene, CDC28, triggers both G1/S and G2/M transitions (Wohlbold and Fisher, 2009). We used the cdc28-as1 allele, which encodes a protein that is specifically inhibited upon addition of the non-toxic ATP analogue, NMPP1 (Bishop et al, 2000; Ubersax et al, 2003). Inactivation of Cdk1 activity correlated with impaired/inefficient processing of DDT-associated SCJ intermediates (Figure 1B). An impairment in the late resolution was also observed when the CLB2 gene encoding for the mitotic cyclin, Clb2, was deleted (Supplementary Figure S1C), albeit to a smaller degree than the one caused by Cdk1 inactivation, likely due to redundancy between the four mitotic cyclins. In all, these results indicate that mitotic Cdk1 activity is crucial for the late resolution.

Because the Plk1/Cdc5 kinase plays central roles in coordinating diverse mitotic events and is activated by Cdc28-mediated phosphorylation (Mortensen et al, 2005), we also examined if Cdc5 activity is required for the late

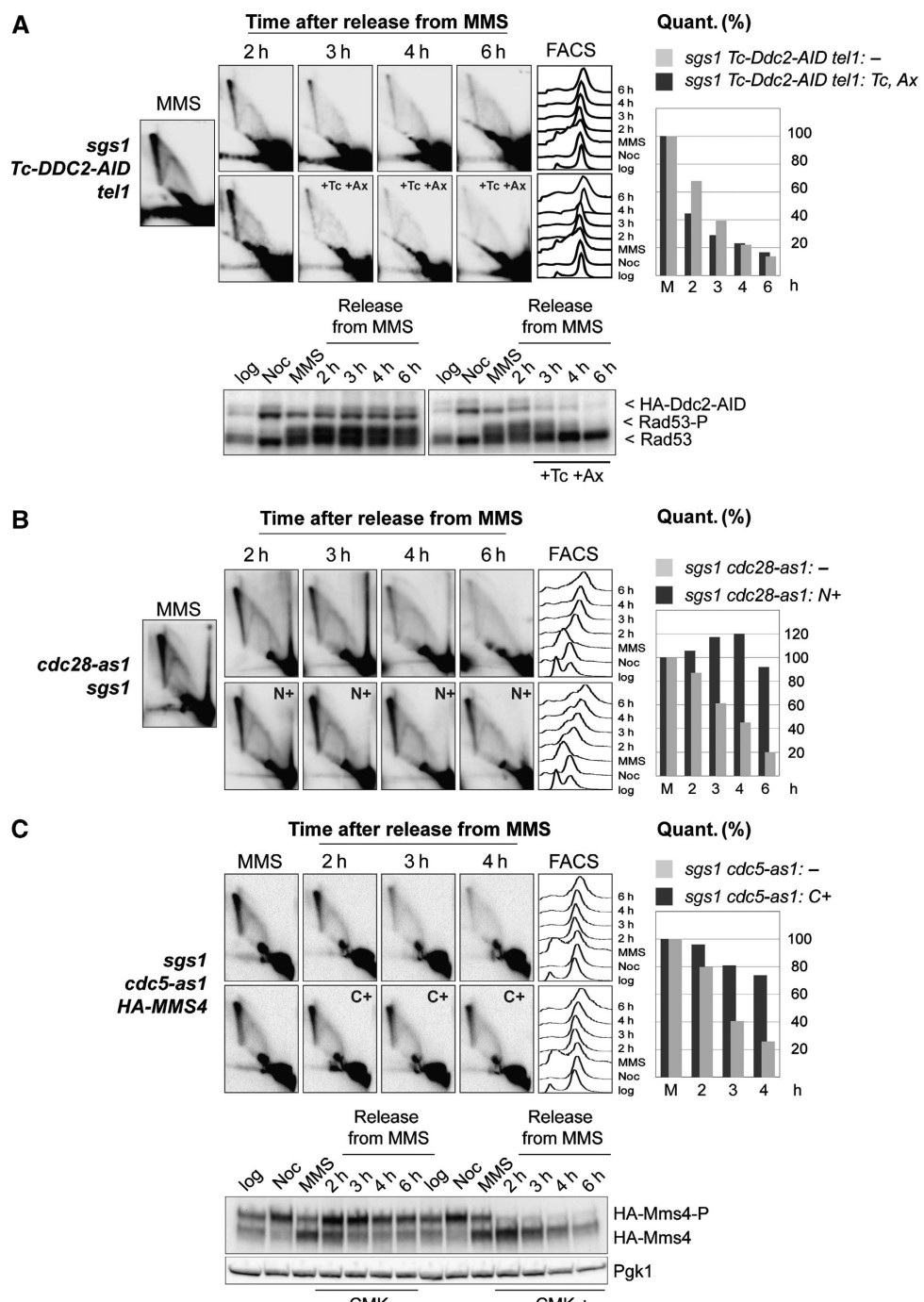

**Figure 1** DNA damage tolerance recombination intermediate resolution in mitosis is damage checkpoint independent, but Cdk1 and Cdc5 dependent. (**A**) The Mec1/Tel1-damage checkpoint is not required for the late resolution. G2/M synchronized *sgs1 Tc-Ddc2-AID tel1* (HY2443) cells were released into MMS for 90 min (MMS), then released into YPD media in two identical sets of cultures. At 2 h following the release from MMS, auxin and tetracycline (Tc, Ax) were added to one set of cultures. Samples were taken at the indicated time points for FACS, protein, and 2D gel analysis. Replication intermediates were visualized using a radioactively labelled probe specific for *ARS305* in 2D gel electrophoresis. The efficiency of checkpoint deactivation was analysed by following Rad53 phosphorylation via immunoblotting. The relative value obtained for the signal of X molecules accumulating after MMS treatment was considered as 100%. (**B**) Cdk1 activity is required for the late resolution. Logarithmically growing *sgs1 cdc28-as1* (HY0666) cells were synchronized in G2 with nocodazole (Noc), released into media containing MMS 0.033% at 30°C for 90 min (MMS), then released into YPD media containing Nocodazole and divided into two identical sets; NMPP1 (N) was added to one set of cultures. At the indicated time points, samples were taken for 2D gel and FACS analysis. The value obtained for X-shaped intermediates in MMS-treated cells was considered as 100% and the other values were normalized to it. (**C**) Cdc5 activity is required for the late resolution. G2/M synchronized *sgs1 cdc5-as1 HA-MMS4* (HY3150) cells were treated with MMS and released into YPD media and divided into two identical sets; CMK-Cl (C+) was added to one set of cultures right after the release, the other set remained untreated. Phosphorylation of Mms4 was analysed by western blot to monitor CMK-Cl efficiency. The relative value obtained for the signal of X molecules accumulating after MMS treatment was considered as 100%.

resolution. We used the *cdc5-as1* allele, which encodes for a protein that is specifically inactivated by chloromethylketone (CMK-Cl) (Snead *et al*, 2007). We monitored the efficiency of CMK-Cl-mediated Cdc5-as1 inhibition by assessing the phosphorylation status of Mms4, a recently reported target of Cdc5 (Matos *et al*, 2011; Gallo-Fernandez *et al*, 2012). Similarly to the effect obtained by inhibiting Cdk1, Cdc5 inactivation drastically impaired the late processing of DDT-associated SCJs (Figure 1C).

Several activities are involved in recombination intermediate processing, such as Mus81-Mms4 and Yen1 nucleases, which are regulated via Cdk1-dependent phosphorylation (Matos *et al*, 2011; Gallo-Fernandez *et al*, 2012). We examined therefore the impact of these factors in the late resolution. As previously reported, we confirmed that Mms4 phosphorylation depends on Cdc28 and Cdc5 activities (Matos *et al*, 2011; Gallo-Fernandez *et al*, 2012) (Figure 1C and see below). Because *mms4Δ* is synthetic lethal in combination with *sgs1Δ* (Mullen *et al*, 2001), in order to address the role of Mus81-Mms4 on the late resolution, we constructed an *MMS4* translational conditional system (Kotter *et al*, 2009), *Tc-MMS4*. Reduced Mms4 levels induced by tetracycline addition correlated with impairment in the late resolution (Supplementary Figure S2A). On the other hand, Yen1, which plays partial overlapping roles with Mus81-Mms4 in DSB recombination intermediate resolution (Ho *et al*, 2010) and is even more potent than Mus81-Mms4 *in vitro* (Matos *et al*, 2011), as well as the Slx1-Slx4 nuclease that also plays functionally overlapping roles with Sgs1/BLM (Mullen *et al*, 2001; Wechsler *et al*, 2011), did not detectably affect the late resolution *in vivo* (Supplementary Figure S2B and C). These latter conclusions are consistent with the ones of (Ashton *et al*, 2011) that used a different experimental set-up. Thus, Cdk1 and Cdc5 are critical regulators of a Mus81-Mms4-dependent DDT intermediate processing pathway that proceeds in G2/M independently of the DNA damage checkpoint.

### Cdk1/Cdc5/Mus81-mediated DDT intermediate processing is restricted to late G2/M when Mms4 phosphorylation potentiates Mus81-Mms4 activity

In line with recent reports, we found that Mms4 phosphorylation proceeds recurrently, in the G2/M of every cell cycle, independently of exogenous DNA damage (Supplementary Figure S3A and (Matos *et al*, 2011; Gallo-Fernandez *et al*, 2012)). To address the impact of the cell cycle-dependent Mms4 phosphorylation on the processing of recombination/DDT intermediates *in vivo*, we constructed an *mms4* allele in which the CDK consensus sites were inactivated (Supplementary Figure S3B). We note that the preferred binding sequence for the Polo-box domain of Cdc5 is overlapping with CDK consensus sites (Elia *et al*, 2003; Lowery *et al*, 2004) and thus Cdc5 may also be involved in the phosphorylation of the selected residues *in vivo* (Matos *et al*, 2011). We identified seven CDK consensus sites in the Mms4 sequence and mutated them to alanine (Ala, A) in order to create an *mms4-7A* allele (Supplementary Figure S3B). The *mms4-7A* allele was then integrated at the *mms4Δ* locus, and its genetic interaction with *sgs1Δ* was examined. The spores isolated for *sgs1Δ mms4-7A* were very slow growing and showed a high degree of lethality, but this synthetic sickness was fully rescued by the *rad51Δ* mutation

(Figure 2A) This result is in discrepancy with the one recently reported by Gallo-Fernandez *et al* (2012), in which mutations of nine CDK consensus sites of Mms4, which include the seven sites we mutated to create the *mms4-7A* allele, did not lead to a growth defect in combination with *sgs1*. We suspect that this may be due to possible strain differences or to the type of *sgs1* mutation employed by the group. Based on the above result, we conclude that the G2/M-specific phosphorylation of Mms4 is required for the processing of persistent, Rad51-dependent DDT recombination intermediates arising in spontaneous conditions.

We further examined the effect of the *mms4-7A* mutation on the late resolution of damage-bypass SCJs. Because of the severe slow growth defect of *sgs1Δ mms4-7A* cells (Figure 2A), we integrated either *Tc-MMS4* or Tc-*mms4-7A* alleles at an ectopic locus in cells in which the wild-type *MMS4* allele was replaced by the conditional allele, *MMS4-AID* (Supplementary Figure S3C). The resulting cells grow normally as the *mms4-7A* mutation is fully complemented by the *MMS4-AID* allele. The *mms4-7A* mutation impaired the late processing of SCJs to an intermediate extent compared to Mms4 depletion (Figure 2B). Similar effects of the *mms4-7A* mutation on DDT-associated X-molecule processing were observed in the *sgs1-D664Δ* background, which is not lethal with *mms4Δ* (Bernstein *et al*, 2009; Supplementary Figure S3D). The observed partial functionality of the Mms4-7A variant may be due to its residual phosphorylation (Figure 2B; Supplementary Figure S3D), as 12 phosphorylation sites on Mms4 were detected via a mass-spectrometry approach (Matos *et al*, 2011). Considering that a decrease in Mms4 phosphorylation leads to Rad51-dependent slow growth defects in combination with *sgs1Δ* (Figure 2A) and to defective processing of persistent SCJs in late G2/M (Figure 2B), we conclude that Mms4 phosphorylation in mitosis potentiates its resolution activity *in vivo*, similarly to what has been proposed *in vitro* (Matos *et al*, 2011; Gallo-Fernandez *et al*, 2012).

Because Mus81-Mms4 can process HJ-like substrates *in vitro* even in the absence of Mms4 phosphorylation, showing the highest activity towards flap and replication fork-like structures (Osman and Whitby, 2007), it remains unknown whether Mus81-Mms4 activity towards DDT intermediates is restricted to G2/M, or whether it occurs constitutively throughout the cell cycle, reaching a peak in G2/M when Mms4 is phosphorylated (Matos *et al*, 2011; Gallo-Fernandez *et al*, 2012). To address this, we examined whether Cdk1/Cdc5- and Mus81-mediated resolution acted also in S phase to resolve damage-bypass SCJ intermediates. Differently from *sgs1Δ*, deletion of either *MMS4* or *MUS81* did not result in an accumulation of X-structures in the proximity of forks (Supplementary Figure S4A; Sollier *et al*, 2009; Ashton *et al*, 2011), and neither did inactivation of Cdk1 (Cdc28-as1) following origin firing (Figure 3A). While *sgs1Δ mms4Δ* lethality was proposed to reflect the outcome of concomitant ablation of two recombination intermediate resolution pathways during replication (Kaliraman *et al*, 2001; Fabre *et al*, 2002; Osman and Whitby, 2007), our above results, together with the findings that *sgs1Δ* is synthetic sick with an *mms4* allele encoding an Mms4 variant specifically defective in G2/M phosphorylation (Figure 2A), indicate that Sgs1 and Mus81-Mms4 occupy different temporal windows, with Mus81-Mms4 being

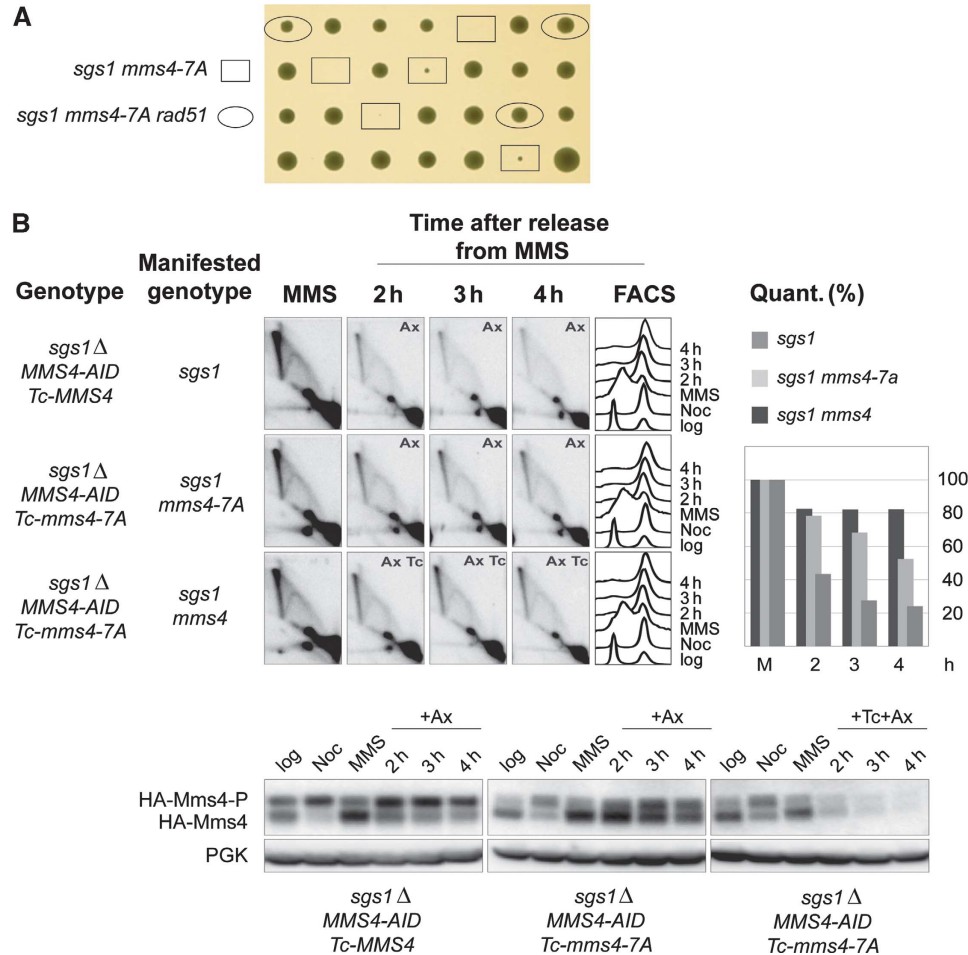

**Figure 2** Mms4 phosphorylation potentiates persistent recombination intermediate processing in G2/M. (**A**) Mms4 essential function in the absence of Sgs1 is related to its role in recombination intermediate resolution in G2/M rather than an overlap of Sgs1 and Mms4 activities in S phase. *sgs1Δ rad51Δ* (HY0992) and *sgs1-D664Δ mms4-7A* (HY2367) strains were crossed. *sgs1 mms4-7A* and *sgs1 mms4-7A rad51* segregants are indicated. (**B**) Cdk1-mediated Mms4 phosphorylation in G2/M is required for late resolution of DDT intermediates. G2 synchronized *sgs1Δ MMS4-AID Tc-MMS4* (HY2640) and *sgs1Δ MMS4-AID Tc-mms4-7A* (HY2642) cells were released into MMS for 90 min (MMS) then released into YPD media containing auxin (Ax). Two identical sets of *sgs1Δ MMS4-AID Tc-mms4-7A* cultures were treated either only with auxin (Ax) to reveal the phenotype of *sgs1 mms4-7A*, or with both auxin (Ax) and tetracycline (Tc) to induce complete Mms4 depletion (*sgs1 mms4*). Samples were taken at the indicated time points for FACS, protein and 2D gel analysis. Depletion and phosphorylation of HA-tagged Mms4 or Mms4-7A proteins were analysed by immunoblotting. The relative values obtained for the X molecules accumulating after MMS treatment were considered as 100%.

restricted to late G2/M. Furthermore, *sgs1Δ cdc28-as1* and *sgs1Δ Tc-MMS4* double mutants behaved similarly with *sgs1Δ* in what regards DDT intermediate accumulation in S phase (Figure 3B; Supplementary Figure S4B).

We conclude that, unlike Sgs1-Top3 or Ubc9/Mms21-dependent SUMOylation (Liberi *et al*, 2005; Branzei *et al*, 2006), Mus81-Mms4 and Cdk1 activities do not play a critical role in DDT intermediate resolution during or soon after completion of replication. Thus, the Cdk1/Cdc5-mediated resolution pathway is likely restrained to late G2/M, or counteracted during replication, rather than just potentiated via Mms4 phosphorylation in mitosis.

### Constitutively active Mus81-Mms4 variants induce deleterious mitotic crossover and replication-associated DDT intermediate processing

To understand the physiological consequence of the cell cycle-regulated Mus81-Mms4 activity, we asked if constitutive activation of Mus81-Mms4 perturbs genome stability. Mus81-Mms4 was proposed to cleave HJ-like intermediates

to generate crossover and non-crossover products in a 1:1 ratio (Schwartz and Heyer, 2011), but so far a contribution of Mus81 to crossover formation in mitotic cells was only revealed following DSB induction (Ho *et al*, 2010), with no effect of Mus81-Mms4 being detected in spontaneous conditions (Robert *et al*, 2006). We asked if artificial activation of Mus81-Mms4 in a cell cycle-deregulated manner would lead to deleterious crossovers, such as those associated with crossover resolution of inter-homologous or inter-heterologous recombination intermediates, known to lead to loss of heterozygosity (LOH) or chromosome translocations (Moynahan and Jasin, 2010).

For this purpose, we constructed phosphorylation-mimicking variants of Mms4 and examined their effect on crossover-associated chromosome translocations using a genetic assay previously described (Robert *et al*, 2006; Supplementary Figure S5A). We mutated a major CDK consensus site on Mms4, S56, which is phosphorylated *in vivo* and contains a perfect Cdc5-binding motif (Matos *et al*, 2011), to glutamic acid (S56E). In addition to the *mms4-S56E* allele, we

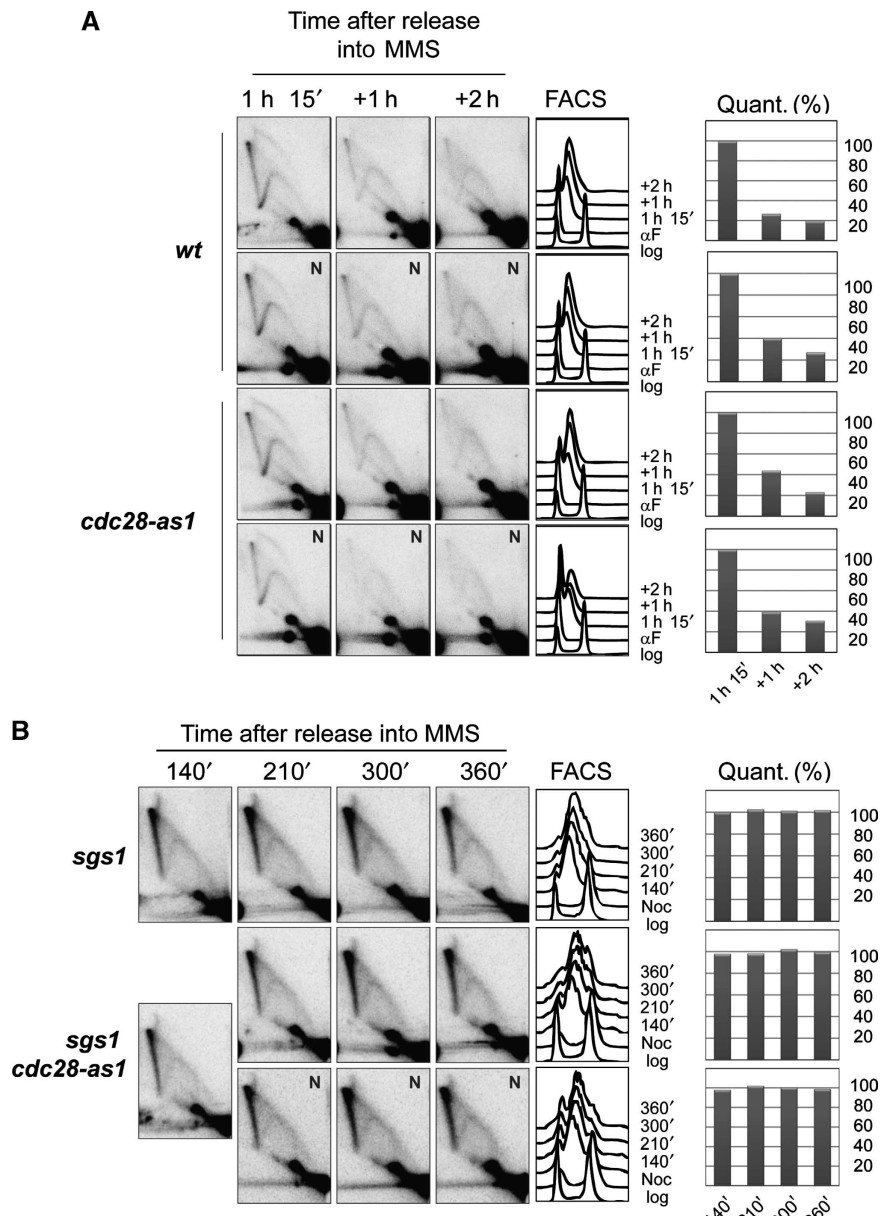

**Figure 3** Cdk1 is not required for damage tolerance intermediate resolution during replication. (**A**) Cdk1 inhibition in S phase following origin firing does not impede damage-bypass intermediate resolution. wt (FY1421) and *cdc28-as1* (FY1422) cells were synchronized in G1, released into MMS for 75 min to allow origin firing, then NMPP1 was added (N) to one of the two identical sets of cultures for each strain. (**B**) Cdk1 inhibition in S phase following origin firing does not exacerbate or reduce the damage-bypass intermediate accumulation in *sgs1* cells. *sgs1* (HY0764) and *sgs1 cdc28-as1* (HY0666) cells were synchronized in G2 and released into MMS containing media. In all, 140 min after the release in the presence of MMS (when most replication origins had fired as judged by the cell morphology), NMPP1 was added to one of two identical sets of *sgs1 cdc28-as1* cultures. (**A**, **B**) Samples were taken at the indicated time points for FACS and 2D gel analysis. For each strain, the highest value obtained during quantification for the X molecule was considered as 100%.

engineered another allele in which besides the S56E mutation, the CDK consensus site, S184, was mutated to aspartic acid (S184D). The *mms4-S56E* and *mms4-S56E, S184D* mutants have wild-type levels of damage sensitivity and the encoded protein variants fully complement the MMS sensitivity of *mms4Δ* cells (Supplementary Figure S5B), as well as the Auxin-induced lethality of *sgs1 MMS4-AID* cells (Supplementary Figure S5C). We found that both Mms4 phosphorylation-mimicking variants boosted spontaneous crossover recombination associated with chromosome translocations (Figure 4A). Furthermore, these variants negatively affected error-free replication across damaged templates, as

observed by a reduction in the X structures representing damage-bypass SCJs (Figure 4B). Thus, temporal deregulation of the Mus81-Mms4 pathway via constitutive Mms4 phosphorylation induces higher levels of deleterious crossover and precocious processing of DDT- and replication-associated SCJs.

### Premature activation of the Cdk1/Cdc5/Mus81-resolution pathway affects replication under genotoxic stress

Because of the negative effects observed with Mms4 constitutive phosphorylation (Figure 4), we hypothesized that regulatory mechanisms may exist in the cell to ensure that

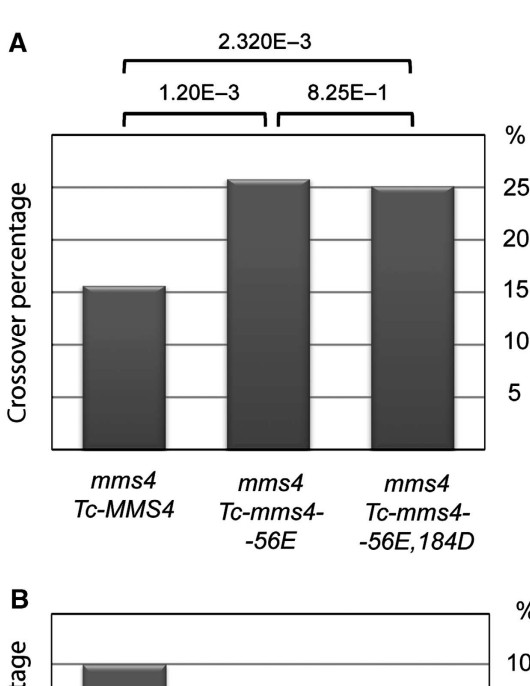

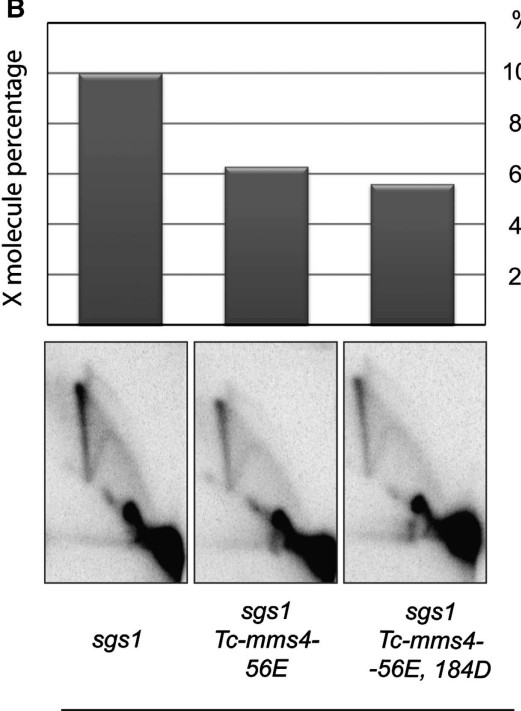

**Figure 4** Constitutively active Mus81-Mms4 variants induce deleterious mitotic crossover and premature processing of DDT intermediates. (**A**) Mms4 phosphomimicking variants induce deleterious mitotic crossover. *mms4Δ Tc-MMS4* (HY2693), *mms4Δ Tc-mms4-56E* (HY3260), and *mms4Δ Tc-mms4-56E, 184D* (HY3264) strains were employed. The strains were streaked for single colonies, eight single colonies from each strain were inoculated into YPD media and grown overnight, then plated on Arg-synthetic media plates for colony development. Spontaneous crossover formation was detected using colony PCR of *ARG+* recombinants, and 192 recombinants were analysed for each strain. *P*-values were determined using 'Chi-Square test' to examine the statistical significance of the differences obtained in the crossover values. (**B**) Mms4 phosphomimicking variants process damage-bypass SCJ intermediates. Logarithmically growing *sgs1Δ* (HY3058), *sgs1Δ Tc-mms4-S56E* (HY3316), and *sgs1Δ Tc-mms4-S56E, T184D* (HY3318) cells were treated with 0.033% MMS at 30°C for 90 min, when samples were collected for 2D gel analyses. The value obtained from *sgs1* samples during quantification for the X molecule was considered as 100%.

the Cdk1-, Mus81-mediated resolution pathway is not activated coincidently with DNA replication. Considering the reported role of the replication checkpoint role in preventing the recruitment of recombination factors at sites of ongoing replication (Meister *et al*, 2005; Barlow and Rothstein, 2009), and several functional interaction between the replication checkpoint Cds1 and Mus81 in *S. pombe* (Kai *et al*, 2005; Froget *et al*, 2008), we asked whether the ATR/Mec1-mediated DNA damage response (DDR) surveillance pathway plays a role in preventing the earlier activation of Mus81-Mms4. By using the conditional *Tc-DDC2-AID* allele described above (see Figure 1A), we found that depletion of Ddc2 during DNA replication resulted in premature activation of the Mus81-Mms4 pathway, as judged by the peaks in Clb2 and Mms4 phosphorylation (Figure 5). We thus validated the Mec1-Ddc2 replication checkpoint mutant/depletion system as a genetic background associated with Cdk1/Cdc5/Mus81 premature activation.

Previous work implicated Mec1 and Rad53 in promoting error-free DDT and template switch intermediate formation (Liberi *et al*, 2005; Gangavarapu *et al*, 2011), but the mechanism involved remained elusive. To investigate if this effect is related to the replication-related function of these checkpoint factors, we first examined the temporal contribution of Mec1-Rad53 to SCJ formation. Conditional depletion of Ddc2 at the beginning of replication, using the *Tc-DDC2-AID* allele described above, led to abolishment of MMS-induced Rad53 activation and a drastic decrease in the DDT-associated SCJs (see Figure 6), whereas checkpoint inactivation later on did not affect the processing kinetics of error-free DDT intermediates or the late resolution (see Figure 1A). These results indicate that the replication-related function of the Mec1-Ddc2 checkpoint is critical for the formation and/or integrity of SCJs mediating error-free DDT.

Considering that Mms4 phosphorylation occurs prematurely when Mec1-Ddc2 function is impaired during replication of damaged templates (Figure 5), we hypothesized that the reduction in template switch intermediates associated with Mec1-Rad53 mutations may reflect an aberrant/precocious processing of the recombination-mediated damage-bypass structures by the prematurely activated Cdk1/Cdc5/Mus81 pathway, rather than a defect in template switch intermediate formation as previously considered (Liberi *et al*, 2005; Vanoli *et al*, 2010; Gangavarapu *et al*, 2011). Strikingly, co-depletion of Ddc2 and of the Mus81-Mms4 endonuclease fully suppressed the SCJ reduction associated with Mec1-Ddc2 inactivation/depletion (Figure 6A). Thus, the defect in error-free DDT attributed to replication checkpoint deficiency is largely due to aberrant processing of SCJs by the prematurely activated Mus81-Mms4 (see Figure 5).

We then investigated the mechanism by which the ATR/Mec1 replication checkpoint counteracts the early action of Mus81-Mms4 in DDT intermediate processing. If Mec1-Ddc2 controls the timing of Cdk1/Cdc5-dependent Mms4 phosphorylation, then inactivation of Cdc5 or Cdc28 should be similar with Mms4 depletion in what regards the SCJ accumulation defect observed in *mec1-ddc2* mutants. On the other hand, if the replication checkpoint regulates directly Mus81-Mms4, or independently of Cdk1 and Cdc5, no rescue of the SCJ reduction associated with checkpoint mutations is to be expected with Cdk1 and Cdc5 inactivation. Inactivation of Cdc5-as1 concomitantly with Ddc2 depletion completely

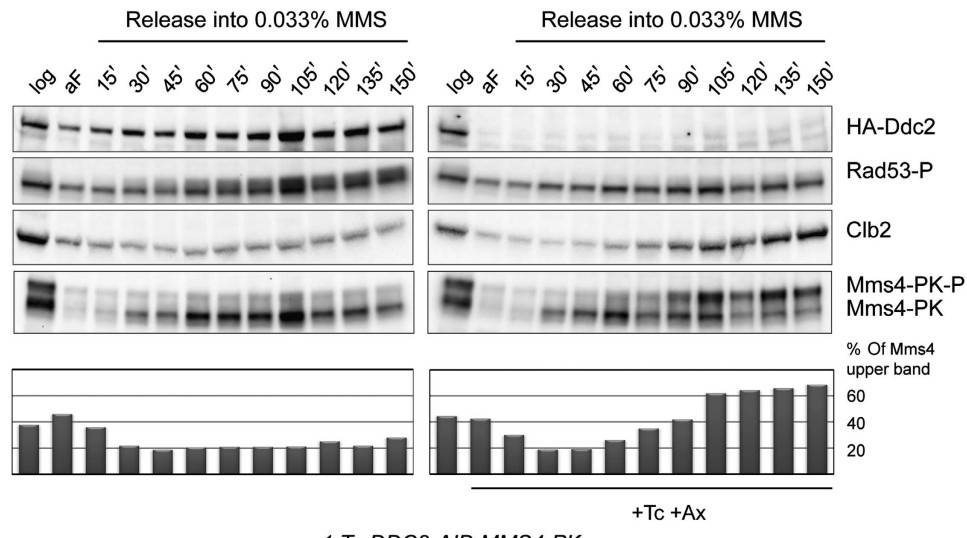

**Figure 5** Premature activation of Cdk1/Cdc5/Mus81 in Mec1-Ddc2-depleted cells. *sgs1Δ Tc-DDC2-AID MMS4-PK* (HY3332) cells were synchronized in G1 phase (aF) and divided into two parts, one half was released into media containing 0.033% MMS, the other half was released into MMS media and treated with both auxin and tetracycline. Samples for protein extraction were collected in the indicated time points. Ddc2 (tagged with 3HA) depletion, Rad53 phosphorylation, Mms4 (tagged with 3HA) phosphorylation, and Clb2 expression levels were analysed by western blot. The percentage of the phosphorylated form of Mms4 was quantified at each time point.

alleviated the decrease in SCJs associated with checkpoint defects (Figure 6B). Similar results were caused by Cdc28-as1 inhibition (Supplementary Figure S6). Notably, no contributions of Cdk1/Cdc5 and Mus81-Mms4 to SCJ processing were observed in replication checkpoint proficient cells in either WT or *sgs1Δ* backgrounds (Figure 3; Supplementary Figure S4), unless constitutively Mms4 variants were employed (Figure 4). Together, these results unexpectedly reveal that the prominent role of the ATR/Mec1 replication checkpoint in error-free DDT is to protect the replication-associated damage-bypass structures from the Mus81-Mms4 nuclease.

### Cdk1/Cdc5-mediated Mms4 phosphorylation contributes to crossover-associated chromosome translocations in checkpoint-deficient cells

We then asked if the premature activation of the Cdk1/Cdc5/Mus81 pathway observed in replication checkpoint-deficient backgrounds (Figure 5) and underlying the error-free DDT defects of these cells (Figure 6) affects genome stability. Various checkpoint mutants reportedly show an increase in genome rearrangements, potentially caused by deleterious crossover resolution of inter-homologue/heterologue recombination intermediates (Robert *et al*, 2006; Moynahan and Jasin, 2010).

If one mechanism by which the S-phase checkpoint suppresses mitotic crossover is via temporal restriction of the Mus81 pathway outside S phase, then the increased crossover outcome associated with replication checkpoint mutations should be to some extent dependent on Mus81-Mms4. We found that *MMS4* deletion partially decreases the crossover outcomes in all S-phase checkpoint mutants analysed: *rad53* (Figure 7A), *mrc1-AQ* (Figure 7B), and *srs2* (Figure 7C). Consistent with previous results, these different checkpoint mutations affect crossover outcomes to varying degrees: *mrc1-AQ* has a modest effect, *rad53* has a stronger increase in crossover, but smaller than *srs2* (Robert *et al*, 2006).

We further addressed the effect of Mms4 phosphorylation on crossovers, taking advantage of the difference between *srs2* and *srs2 mms4* crossover levels (Figure 7C). Integration of the *Tc-MMS4* allele at an ectopic locus in *srs2 mms4* cells fully restored the high levels of crossovers typical of *srs2*, thus confirming that loss of Mms4 function is responsible for the drop in crossovers observed in *srs2 mms4* cells. Notably, the *Tc-mms4-7A* mutation also significantly reduced the crossover increase of *srs2* cells, although to an intermediate effect compared to the one conferred by the *MMS4* deletion (Figure 7C). The intermediate effect of Mms4-7A on crossovers is consistent with its partial effect in the late resolution and may be due to its partial phosphorylation (see Figure 2B). Together with the role of hyperactive Mms4 in inducing crossovers (Figure 4A), these results demonstrate a role for temporally deregulated Mus81-Mms4 in deleterious crossover formation. Furthermore, the correlations between the replication defects and deleterious crossovers induced by premature activation of the Mus81-dependent mitotic resolution pathway (Figures 4, 5, 6 and 7) indicate a new mechanism by which defects in error-free DDT translate into Cdk1-driven genome rearrangements.

## Discussion

Perturbation of the DNA replication process by various cues such as DNA damage and oncogene activation causes 'replication stress', which in turn activates the DDR, a crucial anticancer barrier network mediated by damage and replication checkpoints and their interplay with ubiquitin and SUMO modifications (Bartek and Lukas, 2007; Harper and Elledge, 2007; Polo and Jackson, 2011). Oncogene activation and other types of replication stresses have as common feature the transient accumulation of DNA lesions that presumably drive chromosomal rearrangements and tumorigenesis. Activation of the DNA damage checkpoint kinases ATR-ATRIP and Chk1 (functionally corresponding to

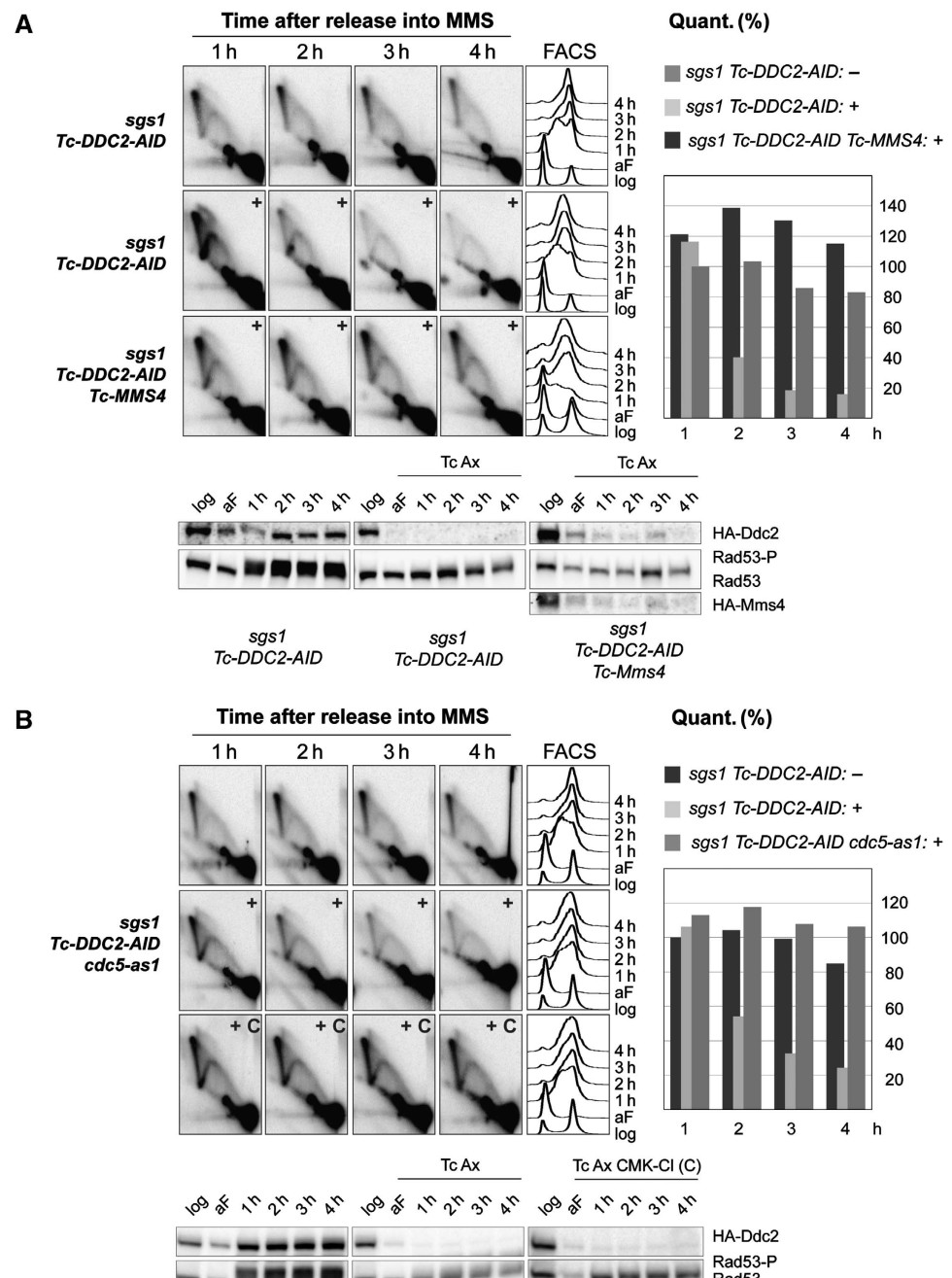

**Figure 6** The replication checkpoint counteracts premature Cdk1/Cdc5/Mus81-mediated processing of SCJs during replication. (**A**) Mus81-Mms4 activity is responsible for the SCJ reduction observed in *ddc2* replication checkpoint-deficient cells. *sgs1Δ Tc-DDC2-AID* (HY2395) cells were synchronized with alpha-factor (aF) and divided into two identical parts. One half of the culture was released into 0.033% MMS without auxin (Ax) and tetracycline (Tc) treatment, the other half was released into 0.033% MMS media containing auxin and tetracycline (+). G1 synchronized *sgs1Δ Tc-DDC2-AID Tc-MMS4* (HY2861) cells were released into 0.033% MMS media containing auxin and tetracycline (+). At the indicated time points, samples were taken for 2D gel and FACS analysis. During quantification, the values obtained for the X molecules accumulating after 1 h MMS release in the untreated series were considered as 100%. The efficiency of Ddc2 and Mms4 depletion was analysed by following HA-Ddc2 and HA-Mms4 expression levels, checkpoint activation was analysed by following Rad53 phosphorylation via immunoblotting. (**B**) Cdc5 activity is responsible for the SCJ reduction observed in *ddc2* replication checkpoint-deficient cells. *sgs1Δ Tc-DDC2-AID cdc5-as1* (HY3451) cells were synchronized with alpha-factor (aF), released into 0.033% MMS containing media, and then divided into three identical sets. One third remained untreated, the second set was treated with auxin and tetracycline (+), and the third part was treated with auxin, tetracycline and CMK-Cl (+C). At the indicated time points, samples were taken for 2D gel and FACS analysis. The values obtained for the X molecules accumulating after 1 h MMS release in the untreated series were considered as 100%. Ddc2 depletion (tagged with 3HA) and checkpoint deactivation were analysed by western blot using anti-HA and anti-Rad53 (EL7) antibodies, respectively.

Mec1-Ddc2 and Rad53, respectively, in budding yeast) constitutes the critical signalling module involved in replication stress response, but the underlying molecular mechanism remains elusive. Our present findings indicate that ATR/Mec1 acts through the replication checkpoint to prevent chromosome rearrangements and to modulate DDT

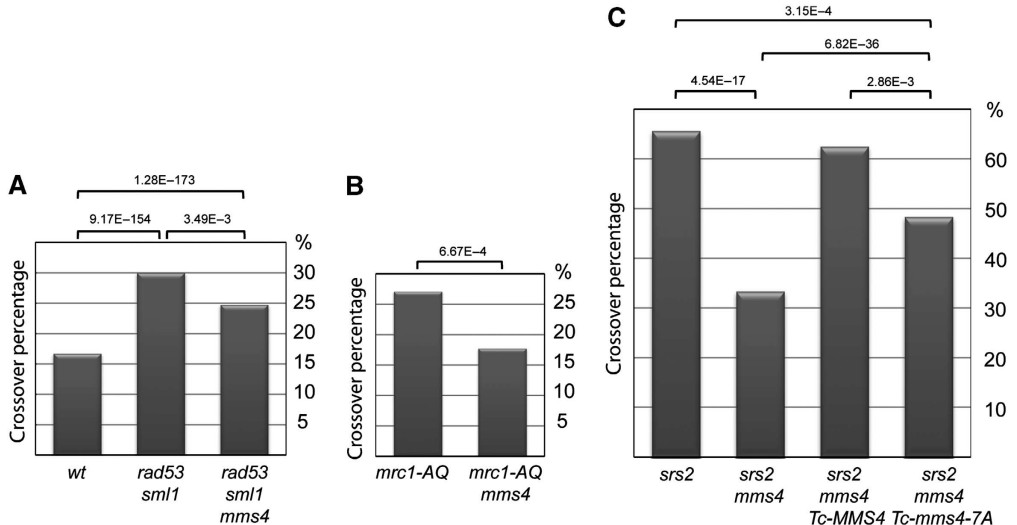

**Figure 7** Cdk1/Cdc5-mediated Mms4 phosphorylation contributes to deleterious crossover in S-phase checkpoint-deficient cells. (**A–C**) Mms4 contributes via its phosphorylation to crossover formation in S-phase checkpoint-deficient cells. wt (FY1485), *rad53Δ sml1* (FY1519), *rad53Δ sml1 mms4Δ* (HY2280), *mrc1-AQ* (FY1518), *mrc1-AQ mms4Δ* (HY3741), *srs2Δ* (FY1521), *srs2Δ mms4Δ* (HY3256), *srs2Δ mms4Δ Tc-MMS4* (HY3247), and *srs2Δ mms4Δ Tc-mms4-7A* (HY3250) strains were used for crossover assay. Eight single colonies from each strain were inoculated into YPD media and grown overnight, then plated on Arg-synthetic media plates for colony development. Spontaneous crossover formation was detected using colony PCR of *ARG*+ recombinants, 96 recombinants were analysed for each strain. *P*-values were determined using 'Chi-Square test' to examine the statistical significance of the differences obtained in the crossover values.

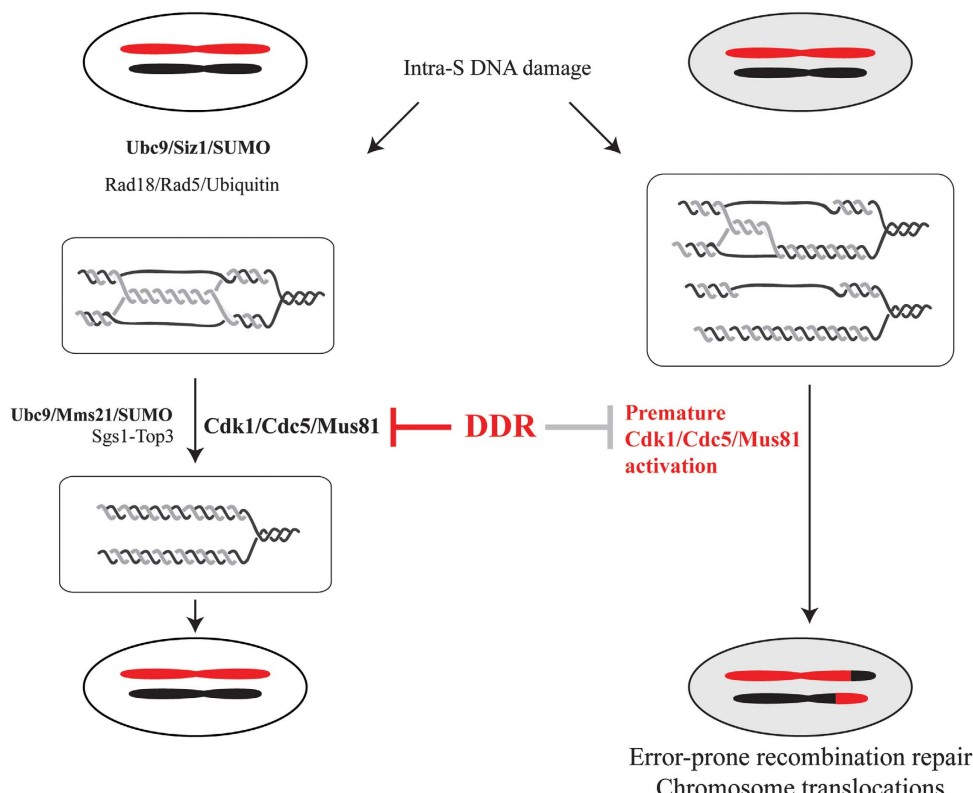

**Figure 8** Temporal restriction of the Cdk1/Cdc5/Mus81 pathway outside S phase is important for genome integrity. Following genotoxic stress, SUMO- and ubiquitin-modification drive error-free DNA damage tolerance by potentiating the formation and resolution of damage-bypass sister chromatid junctions in the proximity of replication forks. In checkpoint-deficient cells, premature activation of the Cdk1/Cdc5/Mus81 pathway induces precocious processing of damage-bypass SCJs, thereby increasing the chances both for faulty template switching and for resolution with crossover outcome. While in physiological conditions the Cdk1/Cdc5/Mus81 pathway acts in G2/M on junctions that primarily involved the sister chromatid and whose processing with or without crossover is genetically silent, the premature activation of this pathway by oncogene activation or DDR/DDT mutations contributes to the crossover-associated genome rearrangements characteristic of cancer cells.

outcome by counteracting the precocious activation of a cell cycle, mitotic CDK1-driven recombination pathway prone to crossover formation (Figure 8).

Recombination template and pathway choices have important implications for genome integrity (Weinert *et al*, 2009). While CDK1 promotes error-free HR-mediated repair of DSBs

(Aylon *et al*, 2004; Ira *et al*, 2004; Jazayeri *et al*, 2006), reports that Cdk1 allows crossover outcomes and is required for formation or recovery of gross chromosomal rearrangements (Enserink *et al*, 2009; Trovesi *et al*, 2011), indicated that Cdk1 must control additional steps associated with error-prone homologous recombination, the nature of which remained to be elucidated. Here, we identified a new error-prone function for Cdk1 in recombination and DDT, carried out in conjunction with the Plk1/Cdc5 kinase and involving the Mus81–Mms4 endonuclease (Figure 8). Importantly, it is the premature activation of this pathway that induces deleterious crossovers (Figures 4, 5, and 7), as its action in G2/M on mitotic recombination intermediates, most of which would have involved the sister chromatid, is expected to be genetically silent.

Our findings imply that, differently from Sgs1-Top3-mediated dissolution that proceeds from S through G2/M (Liberi *et al*, 2005; Karras and Jentsch, 2010), the Cdk1/Cdc5/Mus81-Mms4 resolution pathway is not active during replication (Figures 2A and 3; Supplementary Figure S4). The fact that the Cdk1/Cdc5/Mus81 pathway is driven by mitotic kinases and is not affected by the damage checkpoint-mediated DDR (Figure 1), together with its crossover-prone nature (Figures 4 and 7), suggests that this late resolution pathway acts as a last resort option to allow proliferation, resembling the phenomenon of damage checkpoint adaptation in the face of persistent or overwhelming damage (Pellicioli *et al*, 2001). Thus, by allowing a small window towards mutagenesis and genome rearrangements, which in long term could be advantageous to both cancer cells and evolution, the Cdk1/Cdc5/Mus81-Mms4 pathway protects against more severe loss of information, which in short term would endanger cellular life.

The negative impact of prematurely activating the Mus81-Mms4 resolution pathway highlights the importance for genome integrity of delaying error-prone recombination intermediate processing pathways to G2/M while making space for error-free resolution of DDT intermediates during replication (Figures 4–7). Previous work showed that the highly conserved eukaryotic DDT pathway activated by genotoxic stress during replication involves a task force between ubiquitin- and SUMO-dependent activities to direct the formation and resolution of damage-bypass replication and recombination structures coincidently with DNA replication (Branzei *et al*, 2006, 2008). Our new results underscore the importance of integrating replication-associated error-free DDT activities with cell-cycle driven, last resort options of DDT intermediate processing, explaining the requirement for Cdk1 activity in the context of error-free DDT debilitating mutations (Enserink *et al*, 2009) and revealing the crucial regulatory role of the replication checkpoint in error-free DDT: restraining mitotic Cdk1 activity during replication and relaying DDT control to mitotic Cdk1 in the presence of persistent or late DNA lesions (Figure 8).

In addition to explaining a mechanistic link between error-free DDT defects and tumorigenesis, our work indicates that premature, nuclease dependent processing of the replication-associated junctions involved in damage bypass under conditions of replication and genotoxic stress may lead to a similar outcome with the one induced by oncogene activation: genome rearrangements. By extrapolation, we speculate that temporal deregulation of mitotic Cdk1 activity in general, and the premature activation Cdk1/Cdc5-mediated Mus81-dependent mitotic recombination pathway in particular, may underlie the DDR defects associated with other tumorigenic processes such as oncogene activation, cell cycle, or DDT mutations. Failure of DDR to restrain mitotic Cdk1-driven breakage or recombination until stalled replication intermediates are resolved or until replication or filling of DNA gaps is complete, may be a general mechanism involved in accelerating genomic rearrangements and malignant transformation. Finally, considering that many anticancer drugs act by damaging DNA, our results have therapeutic implications on error-free DDT and checkpoint-defective tumours.

## Materials and methods

### Strains and plasmids
The yeast strains used in this study are derivatives of W303 or DF5 and the relevant genotypes are shown in Supplementary Table S1. Yeast techniques, growing, and synchronization procedures as well as construction of various *mms4* alleles are described in Supplementary Experimental Procedures.

### Protein techniques
The protein extraction procedure is described in Supplementary Experimental Procedures. Rad53 was detected with the mouse monoclonal EL7 antibody (a gift from A Pellicioli). HA-tagged proteins were detected using the anti-HA monoclonal antibody (12CA5 hybridoma at 5 μg/ml dilution). Pgk1, used as a loading control, was detected with the 22C5 monoclonal antibody from Invitrogen (A-6457) using a 2 μg/ml dilution.

### 2D gel analysis and quantification of replication/ recombination intermediates
Each experiment was conducted independently at least twice with qualitatively identical results. If not otherwise indicated, the DNA samples were digested with *Hin*dIII and *Eco*RV and analysed with probes for *ARS305* (Supplementary Figure S1). Purification of DNA intermediates and the 2D gel procedure were carried out as previously described (Lopes *et al*, 2003) with minor modifications. In all, 200 ml cultures ($2-4 \times 10^9$ cells) were arrested by addition of 0.1% sodium azide (final concentration) and cooled down in ice. Cells were harvested by centrifugation, washed in cold water, and incubated in spheroplasting buffer (1 M sorbitol, 100 mM EDTA (pH 8.0), 0.1% β-mercaptoethanol, and 50 U zymoliase/ml) for 1.5 h at 30°C. In all, 2 ml water, 200 μl RNase A (10 mg/ml), and 2.5 ml Solution I (2% w/v cetyl-trimethyl-ammonium-bromide (CTAB), 1.4 M NaCl, 100 mM Tris–HCl (pH 7.6), and 25 mM EDTA (pH 8.0)) were sequentially added to the spheroplast pellets and samples were incubated for 30 min at 50°C. In all, 200 μl Proteinase K (20 mg/ml) was then added and the incubation was prolonged at 50°C for 1 h 30 min, and at 30°C overnight. The sample was then centrifuged at 4000 r.p.m. for 10 min: the cellular debris pellet was kept for further extraction, while the supernatant was extracted with 2.5 ml chloroform/isoamylalcohol (24/1) and the DNA in the upper phase was precipitated by addition of 2 volumes Solution II (1% w/v CTAB, 50 mM Tris–HCl (pH 7.6), and 10 mM EDTA) and centrifugation at 8500 r.p.m. for 10 min. The pellet was resuspended in 2 ml Solution III (1.4 M NaCl, 10 mM Tris–HCl (pH 7.6), and 1 mM EDTA). Residual DNA in the cellular debris pellet was also extracted by resuspension in 2 ml Solution III and incubation at 50°C for 30 min, followed by extraction in 1 ml chloroform/isoamylalcohol (24/1). The upper phase was pooled together with the main DNA prep. Total DNA was then precipitated with 1 volume isopropanol, washed with 70% ethanol, air dried, and finally resuspended in TE 1X. Quantification of X-shaped intermediate signals was performed using the Image Quant software as previously described (Liberi *et al*, 2005; Branzei *et al*, 2008; Vanoli *et al*, 2010). For each time point, areas corresponding to the monomer spot (M), the X-spike signal and a region without any replication intermediates as background reference were selected and the signal intensities (SIs) in percentage of each signal were obtained. The values for the X and monomer were corrected by subtracting from the SI value the background value after the latter was multiplied for the ratio

between the dimension of the area for the intermediate of interest and for background. Thus, the values for $X$ and $M$ were calculated in the following way:

Value for $X = SI$ (Xs) − (SI (background) (area (Xs)/area (background));

Value for $M = SI$ $(M)$ − (SI (background) (area $(M)$/area (background)).

The relative SI for the $X$ was then determined by dividing the value for $X$ with the sum of the total signals (the sum of the $X$ and monomer values). The resulting values for $X$ signals were then normalized as described in the text. For instance, for recovery experiments the relative value of $X$ obtained after MMS treatment was considered as 100% and the other X values were normalized to it.

### Crossover assays

The crossover assays were performed largely as described in Robert *et al* (2006), except that crossover (CO) recombinants were differentiated from non-crossover (NCO) ones by PCR as described in detail below. Cells from each genotype of interest were streaked onto YPD plates for 2 days at 30°C to get individual colonies. Eight or twelve individual colonies from each strain were inoculated independently into YPD medium and incubated for 16 h at 30°C. In all, $10^7$ cells from each eight individual cultures were plated on complete medium lacking arginine to select for recombinants. After 4 days of incubation at 30°C, colonies appeared, each colony corresponding to an independent recombination event. We used a molecular approach to determine if the conversion event is associated with CO. Briefly, the region around *ARG4* was amplified by PCR using three primers in the mixture: xv154 (ACTGGGCGATATGAATGATCC), xv156 (GCCGGAAGCGAGAAGAATC), and xv157 (CTAATTATTCATTGATT TATTCAAGAATTAGC). In a wild-type or NCO configuration, a 2785-bp PCR product is amplified from xv157 that hybridizes upstream from the homology region and xv154 that hybridizes downstream from the homology region on the chromosome VIII. If a CO event occurs, then the reciprocal translocation permits the amplification of a 2204-bp PCR product from the same xv157 primer and xv156 inside the pNM20 plasmid integrated in the chromosome V.

PCR products were resolved by electrophoresis on a 0.8% agarose gel in TAE buffer. In this way for each strain, at least 8*12 (96) PCRs were carried out. For strains with lower, wild-type levels of CO frequencies, such as those shown in Figure 6A, a total of 8*24 (192) PCRs were carried out. CO ratios were calculated by dividing the number of CO events by the number of the sum of CO and non-CO events. To assess the statistical significance of the differences in the crossover ratios, *P*-values were calculated from the number of CO and non-CO events using 'Chi-Square test' (CHITEST) function of the Excel program.

### Supplementary data

Supplementary data are available at *The EMBO Journal* Online (http://www.embojournal.org).

### Acknowledgements

We thank M Foiani, MP Longhese, A Pellicioli, R Visintin, S West, S Gangloff, L Maloisel, and X Veaute for yeast strains, plasmids, antibodies, or experimental protocols, O Branzei for suggestions in presenting the results, and lab members for various help. This work was supported by the ERC grant REPSUBREP 242928, the AIRC grant IG 10637, the Telethon grant GGP12160 and by FIRC.

*Author contributions*: The experiments were designed, conceived, and executed by DB and BS. BS executed most of the experiments. Both authors contributed to strain construction, data analysis and wrote the paper.

### Conflict of interest

The authors declare that they have no conflict of interest.

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
