## [Review Process File · The EMBO Journal]

Manuscript EMBO-2013-84688

Premature Cdk1/Cdc5/Mus81 pathway activation induces aberrant replication and deleterious crossover

Authors: Barnabas Szakal, Dana Branzei

Corresponding author: Dana Branzei, IFOM FIRC Institute of Molecular Oncology

Review timeline:

Submission date:	04 February 2013
Editorial Decision:	01 March 2013
Revision received:	01 March 2013
Accepted:	04 March 2013

Transaction Report:

1st Editorial Decision

01 March 2013

Thank you for submitting your manuscript on Mus81-Mms4 timing control for our consideration. It has now been seen by three expert referees, and I am pleased to inform you that all of them appreciate especially the novel physiological aspects of your work, as well as the study's technical quality, regarding which there were no major concerns (please find the detailed reports copied below). In light of this input, we shall be happy to accept the manuscript for publication in The EMBO Journal, following minor revisions of a few textual and presentational issues raised by referees 2 and 3. When making these modifications, please also make sure to address two editorial points: adding a brief 'Author contribution' description at the end of the text (next to the Acknowledgement/Conflict section), and expanding the currently overly terse 'Material & Methods' section in the main manuscript by moving some of the supplementary methods into the main paper. In my view, it would be relevant to have the last two sections of the 'extended experimental procedures' - on 2D gel analyses and on crossover assays - in the main manuscript.

I am thus returning the manuscript to you for a round of minor revisions, to allow you to make those modifications. Once we will have received your resubmitted final version, we should then be able to swiftly proceed with acceptance and production of the article. This can be further expedited if you at that stage also already send us the completed and signed 'License to publish' and 'Page charge authorization' forms (see below).

Thank you again for the opportunity to consider this work for publication. I look forward to receiving your final version!

REFEREE REPORTS:

Referee #1 (Remarks to the Author):

This is a very interesting and well performed and presented study, clearly relevant to the scope of the EMBO Journal. The corresponding author published previously several important studies on related topics, and the current manuscript goes very significantly beyond that previous work, indeed beyond the state of the art in the field. Overall, the study is based on solid hypothesis, experiments well executed and carefully controlled, the data is solid and conclusions justified. I suggest to accept this work and publish it with a rather high priority, as it could have a stimulating impact on several fields of biology and bio-medicine.

Referee #2 (Remarks to the Author):

The error-free DNA damage tolerance mechanism involves template switching at ssDNA gaps formed during replication and resolution of the resulting sister chromatid junctions (SCJ) by Sgs1-Top3-Rmi1. Hence, SCJs accumulate during MMS treatment of replicating *sgs1* cells. Here, the authors investigate the role of the DNA damage checkpoint, Cdk1 and Cdc5 kinases and Mus81-Mms4 in the late resolution of the SCJs that accumulate in the absence of Sgs1.

The authors show SCJs are slowly removed in G2/M, 2-4 hr after release from MMS, and late resolution requires Cdk1 and Cdc5 activity, but is independent of Mec1 and Tel1. As the Mus81-Mms4 and Yen1 structure-selective nuclease are regulated by phosphorylation during the cell cycle, an Mms4 derivative mutated at 7 putative Cdk1/Cdc5 sites was tested for interaction with *sgs1*. The *mms4-7A* mutant showed recombination dependent lethality with *sgs1*, as observed for the *mms4* null mutant. Late resolution of SCJs was shown to require Mms4 phosphorylation and to be independent of Yen1 or Slx1. Furthermore, an *mms4* phosphomimic reduced SCJs 90 min after release from MMS relative to the MMS4 strain. To address the physiological consequence of deregulating Mus81-Mms4 activity, the authors used an ectopic recombination assay and found a higher percentage of crossover recombinants for the *mms4-56E* mutant. The *mms4* deletion is also able to suppress the hyper-recombination phenotype of *srs2* and replication checkpoint mutants, *rad53* and *mrc1*. Previous studies have shown reduced accumulation of SCJs in replication checkpoint mutants in the absence of Sgs1, interpreted as role for the checkpoint in template switching. Here, the authors demonstrate that *mms4* deletion or inactivation of Cdc5 is able to restore SCJs to the *sgs1* mutant indicating that premature resolution rather than failure to generate SCJs is responsible for reduced level of SCJs.

Although there is some overlap between the data presented here and previous results from the Hickson, Tercero and West labs, there are several novel findings. First, the use of the *mms4* phosphomimic allele to demonstrate premature activation of Mms4 results in increased ectopic crossovers (translocations). Second, the data clearly demonstrate a temporal separation of Sgs1 and Mus81-Mms4 activities, with Sgs1 functioning during replication to suppress the accumulation of SCJs and Mus81-Mms4 acting at G2-M to remove residual structures prior to anaphase. Third, these studies reveal an important function for the DNA damage checkpoint to delay activation of Mms4-Mus81 and prevent deleterious ectopic crossovers.

Comments:

Figure 4: Do the *mms4* phosphomimic alleles over-ride the requirement for Cdc5 for late resolution of SCJs?

Figure 6A: Is the graph labeled correctly? The dark grey bars seem to better represent the *sgs1* Tc-DDC2-AID Tc-MMS4 strain.

Referee #3 (Remarks to the Author):

In this study Szakal and Branzei have tackled the question of the importance of regulating the Mus81-Mms4 complex for genome stability from a different angle than the one previously

published by the West and Tercero labs. Rather than directly investigating if the phosphorylation of Mms4 modulates the catalytic activity of the complex by using *in vitro* nuclease assays, the authors have monitored the appearance and timing of resolution of DNA recombination intermediates. Their results show that loss of Cdk1 and Cdc5-mediated phosphorylation of Mms4 leads to the accumulation in the cell of recombination intermediates, visualized by the persistence in 2D gel analyses of the so-called X-spike representative of Holliday junction and hemi-catenane structures. These data provide strong *in vivo* support to the conclusions of the West and Tercero studies according to which phosphorylation of Mms4 by Cdk1 and Cdc5 is critical to hyper-activate the catalytic activity, especially its HJ resolvase activity, of the Mus81-Mms4 endonuclease. Through a combination of elegant genetic approaches this study also provides evidence that one of the main roles of the replication checkpoint is to protect the replication-associated damage-bypass structures from Mus81-Mms4 nuclease. This is in line with a previous study by Froget and colleagues that revealed a key role of Cds1 and the S-phase checkpoint in fission yeast in preventing unscheduled processing of replication-associated structures by Mus81-Eme1 (Froget et al. 2008). Several additional reports have also been made recently that support a role of Chk1 and Wee1 in protecting replication forks from a toxic activity of Mus81-Eme1 in human cells (Forment et al. 2011; Dominguez-Kelly et al. 2011; Beck et al. 2012).

The principle originality of this study lies in the fact that the authors convincingly show that premature phosphorylation of Mms4 by Cdk1/Cdc5 during replication results in increased rates of crossovers and chromosome translocations along with unscheduled processing of DNA damage tolerance and replication-associated SCJs.

Although the novelty of some of the findings made in this study may be tamed down by previous reports that reach more or less similar conclusions, none actually tackle as well as this study the question of the importance of a timely regulation of Mus81-Mms4 in terms of genome stability. The data provided by Szakal and Branzei are solid and of remarkable quality and as mentioned above, have been obtained from a different experimental approach. Overall, although not always totally novel or unexpected, the findings made in this study do provide a detailed view on the relative contributions of the DNA damage response and the Cdk1/Cdc5/Mus81 pathway in preventing the premature processing of replication associated structures in S-phase while ensuring that persistent DNA recombination structures such as HJs are actively processed before mitosis.

Specific Comments:

- The *sgs1 mms4-7A* mutant shows very slow growth and a high degree of lethality. Strikingly, this synthetic sickness is suppressed by deleting *Rad51*, indicating that phosphorylation of Mms4 and presumably hyper-activation of the catalytic activity of the Mus81-Mms4 resolvase is critical to process DNA structures formed during homologous recombination. The synthetic sickness of the *sgs1 mms4-7A* mutant is in stark contrast with the results of the Gallo-Fernandez et al. 2012 study where mutating the CDK consensus sites in Mms4 had no consequence on cell growth in an *sgs1* background in absence of exogenous genotoxic stress. This discrepancy needs to be discussed.

Fig1A: legend the lower gels. +Tc, Ax?

Fig1SB: legend the lower gels. +Noc?

Page 7, third paragraph: "such as Mms4 and Yen1 nucleases" needs to be rephrased as Mms4 is not a nuclease. Use Mus81-Mms4 instead of just Mms4.

Page 16 typo second paragraph. Mus81-Mms4 not Mms34

Referee #1 (Remarks to the Author):

This is a very interesting and well performed and presented study, clearly relevant to the scope of the EMBO Journal. The corresponding author published previously several important studies on related topics, and the current manuscript goes very significantly beyond that previous work, indeed beyond the state of the art in the field. Overall, the study is based on solid hypothesis, experiments well executed and carefully controlled, the data is solid and conclusions justified. I suggest to accept this work and publish it with a rather high priority, as it could have a stimulating impact on several fields of biology and bio-medicine.

We are very happy that the reviewer finds our work very interesting, conceptually important and going beyond the state-of-the-art in the field.

Referee #2 (Remarks to the Author):

The error-free DNA damage tolerance mechanism involves template switching at ssDNA gaps formed during replication and resolution of the resulting sister chromatid junctions (SCJ) by Sgs1-Top3-Rmi1. Hence, SCJs accumulate during MMS treatment of replicating sgs1 cells. Here, the authors investigate the role of the DNA damage checkpoint, Cdk1 and Cdc5 kinases and Mus81-Mms4 in the late resolution of the SCJs that accumulate in the absence of Sgs1.

The authors show SCJs are slowly removed in G2/M, 2-4 hr after release from MMS, and late resolution requires Cdk1 and Cdc5 activity, but is independent of Mec1 and Tel1. As the Mus81-Mms4 and Yen1 structure-selective nuclease are regulated by phosphorylation during the cell cycle, an Mms4 derivative mutated at 7 putative Cdk1/Cdc5 sites was tested for interaction with sgs1. The mms4-7A mutant showed recombination dependent lethality with sgs1, as observed for the mms4 null mutant. Late resolution of SCJs was shown to require Mms4 phosphorylation and to be independent of Yen1 or Slx1. Furthermore, an mms4 phosphomimic reduced SCJs 90 min after release from MMS relative to the MMS4 strain. To address the physiological consequence of deregulating Mus81-Mms4 activity, the authors used an ectopic recombination assay and found a higher percentage of crossover recombinants for the mms4-56E mutant. The mms4 deletion is also able to suppress the hyper-recombination phenotype of srs2 and replication checkpoint mutants, rad53 and mrc1. Previous studies have shown reduced accumulation of SCJs in replication checkpoint mutants in the absence of Sgs1, interpreted as role for the checkpoint in template switching. Here, the authors demonstrate that mms4 deletion or inactivation of Cdc5 is able to restore SCJs to the sgs1 mutant indicating that premature resolution rather than failure to generate SCJs is responsible for reduced level of SCJs.

Although there is some overlap between the data presented here and previous results from the Hickson, Tercero and West labs, there are several novel findings. First, the use of the mms4 phosphomimic allele to demonstrate premature activation of Mms4 results in increased ectopic crossovers (translocations). Second, the data clearly demonstrate a temporal separation of Sgs1 and Mus81-Mms4 activities, with Sgs1 functioning during replication to suppress the accumulation of SCJs and Mus81-Mms4 acting at G2-M to remove residual structures prior to anaphase. Third, these studies reveal an important function for the DNA damage checkpoint to delay activation of Mms4-Mus81 and prevent deleterious ectopic crossovers.

We are very happy that the reviewer appreciates the novel and physiological significance of our work.

Comments:

Figure 4: Do the mms4 phosphomimic alleles over-ride the requirement for Cdc5 for late resolution of SCJs?

We plan to address in the future if the *mms4* phosphomimic allele can bypass the requirement for Cdc5 in the late resolution, and to identify which phenotypes of Cdc5 and Cdc28 in recombination or DNA damage tolerance can be overcome by constitutive Mus81-Mms4 activation.

Figure 6A: Is the graph labeled correctly? The dark grey bars seem to better represent the sgs1 Tc-DDC2-AID Tc-MMS4 strain.

The graph was not labeled correctly. We corrected this now and we thank the reviewer for pointing this out to us.

Referee #3 (Remarks to the Author):

In this study Szakal and Branzei have tackled the question of the importance of regulating the Mus81-Mms4 complex for genome stability from a different angle than the one previously published by the West and Tercero labs. Rather than directly investigating if the phosphorylation of Mms4 modulates the catalytic activity of the complex by using in vitro nuclease assays, the authors have monitored the appearance and timing of resolution of DNA recombination intermediates. Their results show that loss of Cdk1 and Cdc5-mediated phosphorylation of Mms4 leads to the accumulation in the cell of recombination intermediates, visualized by the persistence in 2D gel analyses of the so-called X-spike representative of Holliday junction and hemi-catenane structures. These data provide strong in vivo support to the conclusions of the West and Tercero studies according to which phosphorylation of Mms4 by Cdk1 and Cdc5 is critical to hyper-activate the catalytic activity, especially its HJ resolvase activity, of the Mus81-Mms4 endonuclease. Through a combination of elegant genetic approaches this study also provides evidence that one of the main roles of the replication checkpoint is to protect the replication-associated damage-bypass structures from Mus81-Mms4 nuclease. This is in line with a previous study by Froget and colleagues that revealed a key role of Cds1 and the S-phase checkpoint in fission yeast in preventing unscheduled processing of replication-associated structures by Mus81-Eme1 (Froget et al. 2008). Several additional reports have also been made recently that support a role of Chk1 and Wee1 in protecting replication forks from a toxic activity of Mus81-Eme1 in human cells (Forment et al. 2011; Dominguez-Kelly et al. 2011; Beck et al. 2012).

The principle originality of this study lies in the fact that the authors convincingly show that premature phosphorylation of Mms4 by Cdk1/Cdc5 during replication results in increased rates of crossovers and chromosome translocations along with unscheduled processing of DNA damage tolerance and replication-associated SCJs.

Although the novelty of some of the findings made in this study may be tamed down by previous reports that reach more or less similar conclusions, none actually tackle as well as this study the question of the importance of a timely regulation of Mus81-Mms4 in terms of genome stability. The data provided by Szakal and Branzei are solid and of remarkable quality and as mentioned above, have been obtained from a different experimental approach. Overall, although not always totally novel or unexpected, the findings made in this study do provide a detailed view on the relative contributions of the DNA damage response and the Cdk1/Cdc5/Mus81 pathway in preventing the premature processing of replication associated structures in S-phase while ensuring that persistent DNA recombination structures such as HJs are actively processed before mitosis.

We are very happy that the reviewer appreciates the novel aspects of our work.

Specific Comments:

- The sgs1Δ mms4-7A mutant shows very slow growth and a high degree of lethality. Strikingly, this synthetic sickness is suppressed by deleting Rad51, indicating that phosphorylation of Mms4 and presumably hyper-activation of the catalytic activity of the Mus81-Mms4 resolvase is critical to process DNA structures formed during homologous recombination. The synthetic sickness of the sgs1Δ mms4-7A mutant is in stark contrast with the results of the Gallo-Fernandez et al. 2012 study where mutating the CDK consensus sites in Mms4 had no consequence on cell growth in an sgs1Δ background in absence of exogenous genotoxic stress. This discrepancy needs to be discussed.

We now discussed in the text about the discrepancy between our results and the ones recently published by Tercero's group regarding the genetic interaction between *sgs1* and *mms4* alleles carrying mutations in the CDK consensus sites (page 8, last 5 lines of the manuscript). Because our initial *mms4-7A* mutation was constructed in a wt background and confirmed by sequencing before proceeding to other crosses, we suspect that some suppressor might have been picked up if Gallo-Fernandez et al. attempted to introduce their *mms4-np* mutation directly in *sgs1* cells. Alternatively, they may not use a complete knock-out of *sgs1* or there might be strain background differences we are not aware of.

Fig1A: legend the lower gels. +Tc, Ax?

We added Tc and Ax to the lower panel gels.

FigS1B: legend the lower gels. +Noc?

We clarified the addition of Nocodazole (Noc) in the lower panel gels.

Page 7, third paragraph: "such as Mms4 and Yen1 nucleases" needs to be rephrased as Mms4 is not a nuclease. Use Mus81-Mms4 instead of just Mms4.

We corrected this as indicated by the reviewer.

Page 16 typo second paragraph. Mus81-Mms4 not Mms34

We corrected this as indicated by the reviewer.

Acceptance letter

04 March 2013

Thank you for submitting your revised manuscript for our consideration. I am happy to inform you that we have now accepted it for publication in The EMBO Journal.

Thank you again for your contribution to The EMBO Journal and congratulations on a successful publication. I hope you will consider us again in the future for your most exciting work!